# Analysis of International Coexistence Management of Genetically Modified and Non-Genetically Modified Crops

**DOI:** 10.3390/plants14060895

**Published:** 2025-03-13

**Authors:** Caiyue Liu, Youhua Wang, Qiaoling Tang, Ning Li, Zhixing Wang, Tan Tan, Xujing Wang

**Affiliations:** 1Biotechnology Research Institute, Chinese Academy of Agricultural Sciences, Beijing 100081, China; caiyueliu@163.com (C.L.); wangyouhua@caas.cn (Y.W.); tangqiaoling@caas.cn (Q.T.); wangzhixing@caas.cn (Z.W.); 2Development Center of Science and Technology, Ministry of Agriculture and Rural Afairs, Beijing 100176, China; lining@agri.gov.cn

**Keywords:** GM crop, non-GM crop, coexistence, labelling management

## Abstract

The coexistence of genetically modified (GM) and non-GM crops has been a subject of considerable concern, particularly in the context of the extensive utilisation of GM crops. In response to this concern, various countries have devised coexistence strategies that are tailored to their respective national contexts, taking into account economic, political, technological and public acceptability factors. In the context of planting, countries such as the United States and Brazil have adopted a strategy of coexistence management, whereby the responsibility for implementing isolation measures falls upon premium producers. In contrast, the European Union, Japan and other countries that import GM crops have enacted legislation requiring growers to adhere to stringent isolation measures to prevent the mixing of GM and non-GM crops. Internationally, GM products are distinguished by a labelling management system to satisfy the public’s right to know and choose and to realise the coexistence of GM and non-GM during circulation and consumption. When considered in the context of China’s specific national conditions, particularly the prevalence of a small-scale peasant economy, it is recommended that China draw upon the lessons learned from the field coexistence strategies employed in countries that have adopted GM planting. This recommendation involves the refinement and enhancement of existing labelling management practices as well as the formulation of a coexistence management policy that is characterised by cost savings, efficiency gains and robust operational capabilities. The implementation of these measures is expected to foster the commercialisation of GM soybean and maize in China.

## 1. Introduction

Genetic modification technology is widely utilised internationally as a significant frontier in agricultural science and technology. In 2023, 206 million hectares of genetically modified (GM) crops were cultivated in 30 countries worldwide, and a total of 75 countries and territories have authorised the utilisation of GM products. The adoption rates of GM cotton and soybeans were 76% and 72.4%, respectively (Figure 1) [1]. The widespread use of GM crops has made it challenging to avoid the mixing of GM and non-GM crops due to gene drift and seed spillage. This has had implications for the purity and quality of agricultural products and has also led to trade disputes due to the mixing of GM crops. As a result, the coexistence of GM and non-GM crops is a major concern of the international community. In response, various nations have adopted divergent regulatory policies tailored to their respective national contexts, with the aim of facilitating the coexistence of GM and organic as well as non-GM products.

At present, China has approved safety certificates for the production and application of 20 GM maize and seven GM soybeans and has validated 64 GM maize and 17 GM soybean varieties (Appendix A). From 2021 to 2023, the government launched a pilot project for the industrialisation of GM maize and soybean, with the pilot project covering 20 counties in five provinces and districts. The results of the pilot project demonstrated that the transgenic maize and soybean exhibited remarkable traits of resistance to insects and herbicides, with a control effect of more than 90% against lepidopteran pests such as the grass moth and a weed control effect of more than 95%. Furthermore, the transgenic maize and soybean have the potential to enhance yields by 5.6–11.6% [2]. The imminent commercial planting of these GM crops has led to the urgent need to address the coexistence of conventional and GM crops, particularly in China.

This study systematically investigates coexistence management strategies and policies in the following GM growing countries: the United States, Brazil, Canada, the Philippines and South Africa. In addition, GM importing countries (regions)—the European Union and Japan—are also examined. Taking into account China’s national conditions and international practices, this paper proposes that coexistence management strategies should be adopted after the large-scale promotion of GM maize and soybean. The modification of the labelling threshold is clearly proposed to promote the smooth and rapid development of biological breeding in China.

## 2. Definition and Connotations of Transgenic and Non-Transgenic Coexistence

As a pioneering domain within the field of agricultural science and technology, the realm of genetic modification technology has witnessed significant advancements. Throughout the annals of agricultural development, a symbiotic coexistence of diverse production systems has been observed, wherein farmers and seed producers have adeptly cultivated a plethora of product types and pure seeds in parallel [3]. The advent of GM crops has precipitated a marked diversification and intricacy in agricultural production systems, exerting a profound influence on global agricultural trade. Notwithstanding, concerns regarding the safety of GM products persist, and certain consumers maintain an opposition to their use. This in conjunction with variances in consumers’ own needs serve to accentuate and underscore the imperative for the harmonious coexistence of GM and non-GM crops.

The term ‘coexistence’ is used to denote the simultaneous existence of different agricultural production systems, including GM, conventional and organic crops. It is important to note that effective measures must be adopted to avoid mutual influence and to ensure that the production systems do not interfere with each other and operate independently [4]. According to the United States Department of Agriculture (USDA) Advisory Committee on Biotechnology and 21st Century Agriculture, coexistence signifies the concurrent cultivation of conventional, organic, IP and GM crops in accordance with consumer preferences and farmer choices [5]. The European Commission defines coexistence as the capacity of farmers to make practical choices between conventional, organic and GM crops while fulfilling their legal obligations concerning labelling and/or product purity standards [6]. As can be seen from the above definitions, the starting point of coexistence is to ensure the effective segregation of the entire industrial chain of GM and non-GM products, from field production to market distribution. The overarching objective is to empower consumers to select the most suitable product according to their preferences while also addressing the diverse requirements of growers and consumers and ensuring their right to information and choice [7].

The concept of coexistence is not associated with environmental or health safety concerns, and it does not pertain to the economic performance of the crop. Instead, it is more closely linked to ensuring fair market access, respecting consumer preferences, maintaining product value, achieving economic efficiency and quantifying the costs associated with implementing coexistence practices [8,9,10]. Furthermore, the issue of coexistence is also relevant to the prevention or minimisation of international trade disputes arising from the mixing of GM and non-GM products. The commercialisation of GM crops can easily lead to the low-level presence of GM and non-GM products due to chance factors, triggering trade disputes. A survey conducted by the Food and Agriculture Organization (FAO) in 2013 revealed that, over the past two decades, more than 20 countries have experienced incidents related to low levels of mixing, primarily involving maize, rice and soybeans [11,12]. Consequently, the adoption of specific measures to ensure the coexistence of GM and non-GM crops has been identified as a means of effectively mitigating international trade disputes arising from these occurrences.

## 3. International Field Crop Management Strategies for the Coexistence of GM and Non-GM Crops

Prior to the approval of the commercial application of GM crops, all countries undertake a safety evaluation of GM products based on internationally recognised and authoritative safety evaluation standards. Stringent segregation measures are also implemented to ensure the safety of GMO field trials. Subsequent to the sanctioning of the commercial utilisation of GM crops, nations have adopted disparate management strategies in accordance with their respective national conditions, which can be summarised into two categories. The first strategy, adopted by the U.S., Brazil and other GM planting countries, involves the implementation of a field planting management mode suitable for the development of the GM industry. This strategy treats the production of GM crops equally with non-GM crops and aims to facilitate coexistence through industry self-regulation and good planting norms. The second strategy, adopted by the European Union, Japan and other GM importing countries (regions), involves the implementation of strict segregation of GM and non-GM field planting legislation to ensure that the mixing of GM and non-GM crops is avoided in the production process.

### 3.1. Coexistence Management Strategies for Field Cultivation in the U.S, Brazil and Other Countries Growing GM Crops

Following the commercialisation of GM crops in the U.S., these crops are now subject to the same production standards as non-GM crops. There are no legal provisions in place; rather, the role of industry associations and market regulation is pivotal. Segregation is achieved by those who pay the premium, with the implementation of locator maps, planting and buffer zones, third-party certification, cooperative exchanges and order management [13]. The “fence-out” rule in production practices imposes a segregation obligation on growers of organic, non-GM and other crops that receive a premium. Growers are obliged to ensure that GM pollen is excluded from their own growing areas in order to guarantee that the crops they produce meet product quality requirements [14]. In 2013, the U.S. Department of Agriculture’s Agricultural Marketing Service promulgated a regulation prohibiting the use of genetically engineered ingredients and GM crops in organic products. In order to avoid the mixing of GMOs, producers of organic products adhere to the corresponding production requirements in accordance with the Organic Food Production Act (OFPA) and the National Organic Program (NOP) and implement effective measures such as planting segregation zones for segregated planting [14,15]. The United States has been at the forefront of major GM crop cultivation, largely due to the adoption of a planting management model that fosters the development of GMOs. From 1996 to 2018, the cultivation of GM crops resulted in economic benefits amounting to $95.9 billion. In 2023, GM crops accounted for 36.1 percent of the global planted acreage, and the adoption rate of the four major GM crops (maize, soybeans, cotton and oilseed rape) was 90 percent or higher [1,16].

Brazil, the second largest GM grower, cultivates primarily GM crops, such as soybeans, maize and cotton, all of which have adoption rates in excess of 90%. The coexistence management model in Brazil is analogous to that of the U.S., wherein GM crops are treated identically to non-GM crops in terms of production following approval for commercial application. Premium products are subject to segregation measures in order to meet their own product quality requirements [17]. For instance, the price premium for non-GM soybeans in Brazil ranges from $0.43 to $0.54 per kilogram, and segregation from GM soybeans is mandatory during cultivation and throughout the transportation chain, necessitating the use of specialised port handling equipment. Decree No. 6323 of 2007 stipulates that GMOs are prohibited at any stage of the production, processing, storage, distribution and sale of organic products [18]. Furthermore, Brazil has established specific regulations pertaining to GM maize, mandating the segregation of GM and non-GM maize in adjacent regions at a distance of ≥100 m or, alternatively, at a distance of 20 m, with the former being planted in conjunction with at least 10 rows of conventional maize of a similar stature and growth cycle to the GM maize [19].

In Canada, the coexistence of GM and non-GM crops is not subject to government regulation. Producers of conventional or organic crops are responsible for implementing segregation measures to avoid contamination with GM crops. The company provides coexistence advice to non-GM crop growers to minimise mixing. Furthermore, certain companies offer guidelines and recommendations for weed management practices, with a view to improving coexistence between GM and non-GM crops [20].

It is evident that developing countries, such as Argentina, the Philippines and South Africa, have not yet formulated policies, regulations or proposals for the coexistence of GM and non-GM crops, including organic agriculture. GM crops that have been authorised for commercial exploitation are regarded as being as safe as conventional crops, and both adopt the same management model [21,22,23].

### 3.2. Coexistence Field Planting Strategies Adopted by GM Crop Importing Countries (Regions) Such as the EU and Japan

The EU and Japan, as well as other importing countries (regions), have established distinct legislative frameworks governing the coexistence of GM and non-GM crops. These frameworks mandate stringent segregation measures for the cultivation of GM crops. Within the confines of such a stringent coexistence regulatory framework, the cultivation of insect-resistant maize MON810 is currently limited to a small area within the EU, specifically in Portugal and Spain, with 0.17 million and 46,300 hectares dedicated to its cultivation in 2023, respectively [1]. However, the EU is also a significant importer of GM maize and soybeans. In 2023, the EU imported 13.4 million tons of soybeans, of which 89% were GM soybeans, and 20.1 million tons of maize, of which 28% were GM maize [24].

The EU has a specific and detailed legislative document for the coexistence of commercialised GM and non-GM crops. In 2003, the European Commission Guidelines for the Development of National Strategies and Best Practice Options for Ensuring the Coexistence of GM and Non-GM Crops (2003/556/EC) was issued, setting out guidelines for “balanced, equitable and compatible” coexistence. Systems for notification, segregation, labelling, delineation of planting areas, public registration, traceability and compensation for damage to GM crops have been established [6]. In 2010, the EU revised the EU Recommendation on Guidelines for the Establishment of National Co-existence Measures to Avoid the Inadvertent Presence of GM Crops in Conventional and Organic Crops (2010/C 200/01) on the basis of practice. The decision on the implementation of coexistence management is left to the member states, which can decide for themselves whether they wish to grow GM crops in their countries, have the right to define the areas where GM crops are grown and take appropriate measures to avoid unintentional mixing of GM and non-GM products. This ensures that consumers and growers have a choice between conventional, organic and GM products [25]. Currently, only 16 of the 27 member states have notified the European Commission of coexistence legislation [26]. Spain published a first draft of a coexistence statute in 2004 but did not implement it because it was not possible to reach consensus among the parties. Portugal is the only country that has both a coexistence statute and its implementation.

Japan attaches great importance to the coexistence of GM and non-GM crops and has established coexistence regulations. The regulations are divided into six chapters, which specify the definition of coexistence, the purpose of its enactment, its scope of application, the provisions related to open general cultivation, the provisions related to open cultivation tests and the rules and penalties. Notably, the regulations encompass a notification system, a quarantine system, an archiving system, a public registration system and a system of compensation for damages [7].

## 4. Coexistence Management Strategies for the Distribution of GM and Non-GM Products

The labelling of GM products is an internationally accepted strategy for the management of coexistence during the distribution of products. The adoption of the New Food Regulation (258/97) by the European Union in 1997 marked the inception of GM labelling. Presently, over 70 countries and regions have established policies pertaining to the labelling and management of GM products. The international GM labelling system can be categorised into four types, depending on whether it is a mandatory requirement and the scope of products that need to be labelled. The four categories are as follows: voluntary labelling, quantitative and comprehensive mandatory labelling, quantitative mandatory labelling by catalogue and qualitative mandatory labelling by catalogue (Table 1) [27].

### 4.1. Voluntary Labelling

Argentina, Canada and the Philippines have adopted voluntary labelling. Argentina has not passed legislation mandating the labelling of GM foodstuffs. The Argentine Ministry of Agriculture has expressed the opinion that food obtained through biotechnology that is as safe as traditional food should not require specific mandatory labelling [28]. Canada has adopted a voluntary labelling policy for GM products, guided by the Voluntary Labelling and Advertising Standards for GM and Non-GMO Products, which was issued in 2004. Products containing less than five percent GM content can be labelled as non-GM [20]. The Philippines currently has no labelling requirements for GM foods, and the Philippine Food and Drug Administration (PFDA) has stated in its “Draft Guidelines for Labelling Prepackaged Foods Derived from or Containing Ingredients of Modern Biotechnology” that it will not require labelling of GM packaged foods [22].

### 4.2. Quantitative and Comprehensive Mandatory Labelling

A number of countries (regions), including the European Union and Brazil, have adopted a quantitative and comprehensive mandatory labelling system. This system stipulates that products must be labelled if they contain GM ingredients above a set threshold. The European Union pioneered the implementation of such a regulatory framework with the issuance of the New Food Regulation (258/97) in 1997, which mandated the labelling of food products that contain GMOs as processed raw materials [27]. In 2001, the EU adopted the Directive on the International Release of GM Organisms into the Environment (2001/18/EC), which stipulates that labelling of GMOs is required at all stages of placing them on the market, with a threshold for GM labelling set at 1%. The EU legislation 1830/2003 and 1829/2003 stipulate that GM content should be below the threshold of 0.9% [29]. In the context of Brazil, the Biosafety Law and Decree 4680/2003 stipulates that GM labelling is mandatory in instances where the content of GM ingredients in the final product volume exceeds 1% [30].

### 4.3. Quantitative Mandatory Labelling by Catalogue

The U.S, Japan and other countries have implemented quantitative mandatory labelling by catalogue, requiring the labelling of products within the scope of the catalogue according to the threshold value. Japan has mandated the labelling of 33 products made from nine different crops, including soybeans, maize, potatoes, oilseeds, cottonseeds, alfalfa, sugar beets and papayas, when the top three raw materials and 5% or more of the product’s dry weight contain more than 5% GM ingredients [27,31].

Prior to 2016, the U.S. had adopted a voluntary labelling policy. However, on 29 July 2016, then-President Barack Obama signed the National Genetically Engineered Food Disclosure Standard (i.e., S. 764 Act), which clarified that the United States would implement mandatory labelling for bioengineered foods. On 21 December 2018, the U.S. Department of Agriculture (USDA) issued guidance on the National Bioengineered Food Disclosure Standard (NBFDS), which specified that the labelling threshold is set at 5 percent. Concurrently, the USDA’s Agricultural Marketing Service (AMS) has formulated a list of labelling requirements for bioengineered foods, encompassing alfalfa, apples, canola, maize, cotton, eggplant, papaya, pineapple, potatoes, salmon, soybeans, squash and sugar beets [32]. It is important to note that S.764 explicitly mandates labelling without prejudice to the conclusion that GM foods are as safe as conventional non-GM foods and requires labelling only for products with GM ingredients in the end product. The modification of the U.S. labelling policy is not associated with considerations of safety; rather, its objective is to ensure federal uniformity in labelling legislation, to reduce the cost of production and distribution of GM products and to address public demands for mandatory labelling of GM.

### 4.4. Qualitative Mandatory Labelling by Catalogue

Presently, China stands as the sole nation that mandates the implementation of qualitative labelling by catalogue. In accordance with the Measures for the Administration of Labelling of Agricultural GMO and the National Standard for Labelling of Agricultural GMO, qualitative labelling according to the catalogue has been implemented for 17 products of five types of crops, including GM soybeans, maize, oilseed rape, cotton and tomatoes. It is noteworthy that all agricultural GM products included in the labelling management catalogue and utilised for marketing, irrespective of the detectability of GM ingredients in the final product, are obligated to adhere to GM labelling [30].

## 5. Implications of the International Policy on Coexistence Management of GM Crops for China

### 5.1. Coexistence Management with Segregation of Premium Subjects Facilitates Industrial Application of GM Crops

In the context of GM crop cultivation, prominent countries (regions), such as the U.S., Brazil and South Africa, have adopted uniform field management practices for both GM and non-GM crops. The segregation responsibilities are shouldered by producers of premium products. The coexistence of these crops in field cultivation is achieved through market orientation and the implementation of good cultivation practices. From an implementation perspective, this approach has been shown to reduce the cost of planting isolation and to be more conducive to the utilisation of the technical advantages of transgenic cost-saving and efficiency. The United States, a nation with a strong agricultural sector, has attained a favourable position in global trade by virtue of the cost-effectiveness of promoting GM crops. Concurrently, the nation has sustained its pre-eminence in the realm of research and development in transgenic technology [13,17]. Conversely, South Africa and the Philippines, as predominantly smallholder economies, have enhanced the competitiveness of agricultural production and supply security through the promotion and application of transgenic technology [21,22,23].

The EU and Japan, as major importers of GM products, have implemented regulatory frameworks that impose restrictions on the cultivation of GM crops. These measures are designed to safeguard the competitiveness of domestic agricultural production. Separate legislation on the coexistence of GM and non-GM crops imposes stricter regulatory requirements and places the responsibility for segregation on the growers of GM crops [6,7]. While this regulatory framework facilitates the identification of GM products, it concomitantly increases the cost of cultivation and the complexity of regulation. Consequently, no GM crops have been cultivated for application in Japan, and only Portugal and Spain have planted GM crops in the European Union, where the industrialisation process has been gradual and the cultivated area has been diminishing.

At present, the Chinese agricultural sector is characterised by a small-farm economy with small and dispersed fields per household. This has made the segregation of GM crops costly and difficult to implement and regulate. The industrialisation of GM corn and soybeans has been progressively promoted, and favourable supporting planting and production norms have been established. Concurrently, farmers have been empowered to experience the benefits of GM cultivation, and public recognition of GM has been enhanced. In contrast, the coexistence management policy of the United States, Brazil and South Africa, as well as other transgenic planting countries, has been found to impose relatively low segregation costs and to be more operational, more in line with the current situation of China’s agriculture, and conducive to promoting the overall development of China’s agriculture. Furthermore, it has been found to effectively safeguard the supply of soybeans, corn and other agricultural products and other bulk supply needs.

### 5.2. Labelling Management as an Effective Means of Differentiating Between GM and Non-GM Products

In the context of GM products, various countries and regions have adopted labelling management systems with the aim of differentiating between GM and non-GM products. This initiative is driven by the desire to satisfy the public’s right to know and right to choose, as articulated by various stakeholders. The demand for mandatory labelling has been a prominent subject in international discourse. In the United States, in an effort to address consumers’ demand for mandatory labelling, the voluntary labelling of GMOs has been transitioned to quantitative mandatory labelling. Similar calls for mandatory labelling have emerged in Argentina and the Philippines, as evidenced by the Philippine GMO Labelling Act, Right to Know Act’ bill (House Bill No. 6411) introduced in the 18th Philippine Congress in 2020. This bill proposes the mandatory labelling and regulation of GMOs or food products containing GM ingredients and food products produced through genetically engineered technologies [22].

The level of public awareness and acceptance of GM products is a significant factor in the management of labelling. The transition from voluntary to mandatory GM labelling in the U.S. is indicative of the significant impact that consumer preferences have had on the labelling system. The 2016 National Bioengineered Food Disclosure Standard in the U.S. aims to standardise GM food labelling methods and prevent a scenario where states adopt divergent policies, potentially leading to conflict with federal regulations. Japan and the European Union have established mandatory labelling systems that are informed by public acceptance and consumption concerns as well as the need to safeguard national food security. A survey of 434 college students’ choices of 64 foods labelled as GMO-free, non-GMO or GMO found that 85% of the college students believed that GM foods had more or less of an impact on their health but did not influence their choice of GM foods [33].

The present study explores the efficacy of China’s current regulatory framework for GM crops. The regulatory approach in China is qualitative in nature, representing the most stringent form of regulation. However, the effectiveness of this approach in practice is questionable. The existing technology and management systems are not capable of ensuring the complete segregation of GM and non-GM crops. From the perspective of safeguarding consumers’ right to know and right to choose, if products unintentionally mixed with 0.01% of GM ingredients are labelled as GM products in the same way as products containing 5% or higher levels of GM ingredients, the indicative significance of the labelling will be diminished, which will in fact affect consumers’ right to choose.

## 6. Conclusions

In consideration of the present status of R&D and the implementation of bio-breeding in China along with the policy directives aimed at accelerating the industrialisation of bio-breeding, it is evident that the domain of bio-breeding industrialisation in China is undergoing annual expansion. Concurrently, the coexistence of transgenic and non-transgenic genes along with the occurrence of low-level mixing are unavoidable consequences. It is recommended that the experience of GM planting countries, such as the United States, Brazil and South Africa, be taken into consideration, with the aim of learning from their coexistence management experience. The adoption of a coexistence management model in which the segregated subject is the premium or special demand subject and the special demand is met through the order farming mode is proposed. This is considered to be conducive to solving the problem of low-level mixing in trade and to meeting the diversified needs of the market for non-GM foods, organic foods and so on. When considering factors such as testing technology, support capacity and public acceptance of GM products, it is recommended that the labelling threshold be set at 3%. This will help maintain the continuity of regulations, respect public acceptance and avoid excessive labelling of products whose raw materials are unintentionally mixed with trace amounts of GMOs. It is noteworthy that China has approved a total of 51 currently valid safety certificates for plant production and application, of which 41 are GM products (25 for maize, 10 for soya bean, 2 for cotton, 2 for rice, 2 for papaya) and 10 are gene-edited products (2 for maize, 5 for soya bean, 2 for wheat, 1 for rice) [2]. This situation calls for a prompt response in the form of an urgent update to the labelling of crops and product catalogues, taking into account the application of imported and domestic transformants. Such an update is necessary to ensure the protection of the public’s right to information. Additionally, it is essential to establish clear requirements for non-GMO labelling, with the aim of preventing market confusion and ensuring fair competition within the industry.

## Figures and Tables

**Figure 1 plants-14-00895-f001:**
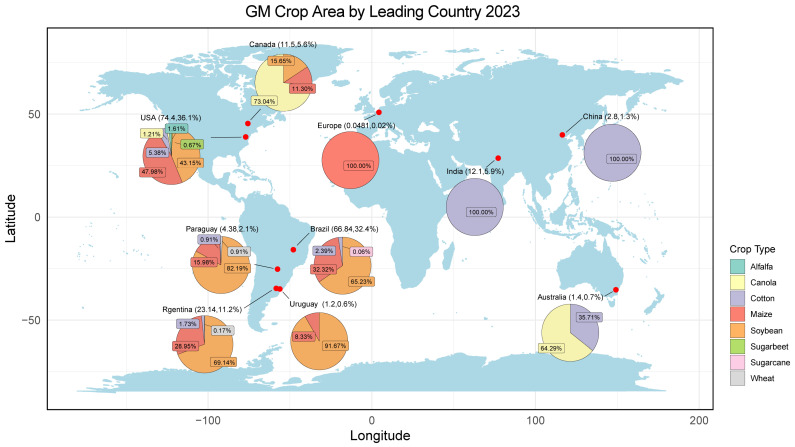
GM crop area by leading country (Ha m.) [1]. The world map was generated using the ‘maps’ package in R.

**Table 1 plants-14-00895-t001:** GM labelling management system in major countries (regions).

Type	Country (Region)	Label Threshold	Label Object
Voluntary labelling	Canada	—	—
Voluntary labelling	Argentina	—	—
Voluntary labelling	Philippines	—	—
Quantitative and comprehensive mandatory labelling	European Union	0.9%	Products with more than 0.9% GM ingredients
Quantitative and comprehensive mandatory labelling	Brazil	1%	Products with more than 1% GM ingredients
Quantitative mandatory labelling by catalogue	United States	5%	Foods containing more than 5% of the 13 GM plants and animals on the list of bioengineered foods
Quantitative mandatory labelling by catalogue	Japan	5%	33 categories of processed foods containing more than 5% of 9 GM crops in the main ingredients
Qualitative mandatory labelling by catalogue	China	0%	5 GM crops and products containing ingredients from these 5 GM crops

Note: Labelling by catalogue means that only products in the catalogue need to be labelled, while other products do not need to be labelled.

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
