# Peer review of "Analysis of International Coexistence Management of Genetically Modified and Non-Genetically Modified Crops"

_plants, 2025, doi:10.3390/plants14060895_

Round 1
Reviewer 1 Report
Comments and Suggestions for Authors
Dear authors and editors,
The paper is topical of the contemporary international conditions and the need of sustainability in the agriculture. As a whole the paper is interesting. I would like to support the authors to improve that paper to become a good one and to be provided to the public. According to this the paper needs some improvements:
1. The paper is not exactly scientific. It the present form it looks like a popular article in agricultural journal. The paper should follow the scientific structure that provides itself the technology of preparing such papers. Firstly, the paper should present the scientific level that has been achieved in the field by previous research. The second part should be dedicated to the methodology. Here the methodology is not clear. After that there should be presented the results and the discussion. In the end should be placed the conclusions. Only after following the above structure will be possible to the reader to outline the scientific contribution of the paper.
2. The paper includes many statements regarding Chinese legislation of the co-existence of GM and non-GM. I see there the great potential of the paper to be revised in this direction and probably the title of the paper to be changed.
3. The most important here is to be defined exact and clear purpose of the paper.
4. There was mentioned that European Union is a country (row 162). Please correct that.
5. There are very big tables in the beginning of the paper. Please provide them like appendixes.
I wish authors a success.
Author Response
Dear editor,
Thanks so much for your work of our manuscript! We had revised our manuscript according to the reviewers’ suggestions, and we hope it will meet the publish standard. Following contents are the response to the review comments.
Reviewer Comments:
Reviewer 1
Reviewer #1: The paper is topical of the contemporary international conditions and the need of sustainability in the agriculture. As a whole the paper is interesting. I would like to support the authors to improve that paper to become a good one and to be provided to the public. According to this the paper needs some improvements:
- The paper is not exactly scientific. It the present form it looks like a popular article in agricultural journal. The paper should follow the scientific structure that provides itself the technology of preparing such papers. Firstly, the paper should present the scientific level that has been achieved in the field by previous research. The second part should be dedicated to the methodology. Here the methodology is not clear. After that there should be presented the results and the discussion. In the end should be placed the conclusions. Only after following the above structure will be possible to the reader to outline the scientific contribution of the paper.
Response: We are grateful to the inquiring party for the query posed. The present article principally analyses the policy discrepancies between international GM and non-GM crop coexistence management, with the objective of providing decision-making support for China's biobreeding industrialisation. Consequently, this article is an academic paper of the nature of a review, which is not entirely consistent with the structure of a research paper. In light of the suggestions provided, the paper's structure has been revised by the addition of an introduction and conclusion section (lines 48 to 107 of the introduction and lines 660 to 686 of the conclusion).
- The paper includes many statements regarding Chinese legislation of the co-existence of GM and non-GM. I see there the great potential of the paper to be revised in this direction and probably the title of the paper to be changed.
Response: We are grateful for the question. Within the conclusion, we have proposed policy recommendations that must be improved to promote the expansion and acceleration of the industrialisation of bio-breeding in China in the context of the present circumstances. We have also made changes to the title, which now reads 'Impact of coexistence management between GM and non-GM crops on industrialization of biological breeding in China'. Should further suggestions for the title be forthcoming, we will engage in a process of discussion and optimisation.
- The most important here is to be defined exact and clear purpose of the paper.
Response: We appreciate the reviewer’s positive evaluation of our work. This article has analysed the coexistence management policies of GM and non-GM crops in the production and distribution and consumption segments in the international arena. The aim of this analysis was to combine China's national conditions with international practices, and to put forward suggestions to optimise and improve the relevant policies. The overarching objective of this analysis was to promote the smooth and rapid development of biological breeding in China (lines100 to 107).
“This study systematically investigates coexistence management strategies and policies in the following GM growing countries: the United States, Brazil, Canada, the Philip-pines and South Africa. In addition, GM importing countries (regions) – the European Union and Japan – were also examined.Taking into account China's national condi-tions and international practices, this paper proposes that coexistence management strategies should be adopted after the large-scale promotion of GM maize and soybean. The modification of the labelling threshold is clearly proposed to promote the smooth and rapid development of biological breeding in China.”
- There was mentioned that European Union is a country (row 162). Please correct that.
Response: We truly appreciate the reviewer for pointing out these details. The error has been corrected and other statements in the article have been checked.
- There are very big tables in the beginning of the paper. Please provide them like appendixes.
Response: We are extremely grateful to reviewer for pointing out this problem. These data have been presented as Supplementary Table 1.
Thank you again for your contribution to our manuscript.

Reviewer 2 Report
Comments and Suggestions for Authors
This article deals with a very interesting question, however the article needs quite a bit of proofreading and I believe that it is a bit unfinished.
The description of a review article from Plants reads:
'Reviews offer a comprehensive analysis of the existing literature within a field of study, identifying current gaps or problems. They should be critical and constructive and provide recommendations for future research.'
However, there is not much of a discussion of the current gaps or problems. More literature could also be cited.
I recommend that the long table of GM varieties in China be put in the appendix. Additionally, the article needs to be reviewed carefully for language.
Figure 1 needs some revision. Instead of a crop, the European circle says 'Portugal' - I am not sure how to interpret this.
I would also recommend a table comparing the different forms of regulation, the countries that follow each type, and their benefits and disadvantages.
Comments on the Quality of English LanguageThere needs to be substantial proofreading and copy-editing of this article to pick up on numerous typos (e.g., 'wildly used' in the abstract; Figure 1 'corp area' instead of 'crop area;' lines 59-60, the sentence trails off, many others).
Author Response
Dear editor,
Thanks so much for your work of our manuscript! We had revised our manuscript according to the reviewers’ suggestions, and we hope it will meet the publish standard. Following contents are the response to the review comments.
Reviewer Comments:
Reviewer 2
Reviewer #2: This article deals with a very interesting question, however the article needs quite a bit of proofreading and I believe that it is a bit unfinished. The description of a review article from Plants reads:
- Reviews offer a comprehensive analysis of the existing literature within a field of study, identifying current gaps or problems. They should be critical and constructive and provide recommendations for future research. However, there is not much of a discussion of the current gaps or problems. More literature could also be cited.
Response: We are grateful to the inquiring party for the query posed. In light of the aforementioned recommendations, an introduction and a conclusion have been incorporated into the paper. The conclusion section of the paper contains recommendations regarding the systems that need to be optimised in the process of advancing the industrialisation of biobreeding. These recommendations are based on the research and application of transformants in China and the policy direction (lines 48 to 107 of the introduction and lines 660 to 686 of the conclusion).
- I recommend that the long table of GM varieties in China be put in the appendix. Additionally, the article needs to be reviewed carefully for language.
Response: We are very grateful to the reviewer for pointing this out. These data have been presented as Supplementary Table 1.
- Figure 1 needs some revision. Instead of a crop, the European circle says 'Portugal' - I am not sure how to interpret this.
Response: We truly appreciate the reviewer for pointing out these details. The error has now been rectified and the image has been re-uploaded (line 90).
- I would also recommend a table comparing the different forms of regulation, the countries that follow each type, and their benefits and disadvantages.
Response: We appreciate the reviewer’s positive evaluation of our work. A comparison was made of the differences in labelling thresholds and labelling objects of labelling management policies in major countries (regions), the results of which are set out in Table 1 (line 402).
- There needs to be substantial proofreading and copy-editing of this article to pick up on numerous typos (e.g., 'wildly used' in the abstract; Figure 1 'corp area' instead of 'crop area;' lines 59-60, the sentence trails off, many others).
Response: We are extremely grateful to reviewer for pointing out this problem. These errors have been corrected and other statements in the article have been checked.
Thank you again for your contribution to our manuscript.

Round 2
Reviewer 1 Report
Comments and Suggestions for Authors
Dear Editors,
As I saw the manuscript after revisions, it is significantly improved. The Authors patiently adressed all the remarks after the first review. I consider the current manuscript like fully prepared and ready to be published.
Author Response
Thank you again for all your work on our manuscripts!
Reviewer 2 Report
Comments and Suggestions for Authors
Thank you, I am satisfied with your edits that have helped to strengthen the article greatly.
The only thing remaining: the title of Table 1 is now listed as 'This is a table. GM labelling management system in major countries (regions).'
I believe the first statement 'This is a table' should be taken out.
Author Response
Comments 1:The only thing remaining: the title of Table 1 is now listed as 'This is a table. GM labelling management system in major countries (regions).'
I believe the first statement 'This is a table' should be taken out.
Response 1:We sincerely thank the auditor for pointing this out. The error has now been corrected (line 402).
Thank you again for all your work on our manuscripts!